# Towards Better Generalization via Distributional Input Projection Network

## Abstract

As overparameterized models become increasingly prevalent, training loss alone offers limited insight into generalization performance. While smoothness has been linked to improved generalization across various settings, directly enforcing smoothness in neural networks remains challenging. To address this, we introduce Distributional Input Projection Networks (DIPNet), a novel framework that projects inputs into learnable distributions at each layer. This distributional representation induces a smoother loss landscape with respect to the input, promoting better generalization. We provide theoretical analysis showing that DIPNet reduces both local smoothness measures and the Lipschitz constant of the network, contributing to improved generalization performance. Empirically, we validate DIPNet across a wide range of architectures and tasks, including Vision Transformers (ViTs), Large Language Models (LLMs), ResNet and MLPs. Our method consistently enhances test performance under standard settings, adversarial attacks, out-of-distribution inputs, and reasoning benchmarks. We demonstrate that the proposed input projection strategy can be seamlessly integrated into existing models, providing a general and effective approach for boosting generalization performance in modern deep learning.

## 1 Introduction

Overparameterization has become a defining feature of modern deep learning. From vision transformers to large language models, today's networks often possess far more parameters than training examples. While this overparameterization enables remarkable expressivity and optimization ease, it is of great importance to identify models with strong generalization performance (i.e, the excellent performance on unseen test data).

Recent work has sought to characterize generalization through various geometric and functional properties of the learned models. Notably, Johnson & Zhang (2023) argue that the generalization gap can be largely attributed to two components: inconsistency and instability, with the latter being closely related to the Lipschitz continuity of the learned function. In parallel, another prominent line of research focuses on sharpness, typically quantified via the spectral norm of the Hessian. Building on this perspective, Foret et al. (2020) proposed Sharpness-Aware Minimization (SAM), a technique aimed at locating flatter regions of the loss landscape. SAM has been successfully applied in diverse domains including computer vision (Chen et al., 2021), natural language processing (Bahri et al., 2021), and bi-level optimization (Abbas et al., 2022).

However, enforcing low sharpness, or low Lipschitz constants during training remains a significant challenge—particularly without sacrificing generalization. For instance, there is no guarantee that a model with low Lipchitz on the training distribution will remain a low Lipschitz on the test distribution. Adversarial training (Madry et al., 2017) and Random Smoothing (Cohen et al., 2019) have proven effective in reducing the Lipschitz constant and improving robustness, but often introduces a trade-off between robustness and standard generalization performance (Tsipras et al., 2018; Zhang et al., 2019; Javanmard et al., 2020; Donhauser et al., 2021; Dobriban et al., 2023; Hao & Zhang, 2024). Likewise, while SAM can find flatter regions of the landscape, it does not always yield true minimizers, even in sufficiently flat neighborhoods (Yue et al., 2023).

An increasingly promising direction involves promoting smoothness in the learned models. Some recent works (Johnson & Zhang, 2023; Shen & Meinshausen, 2023) suggest that smoother input-

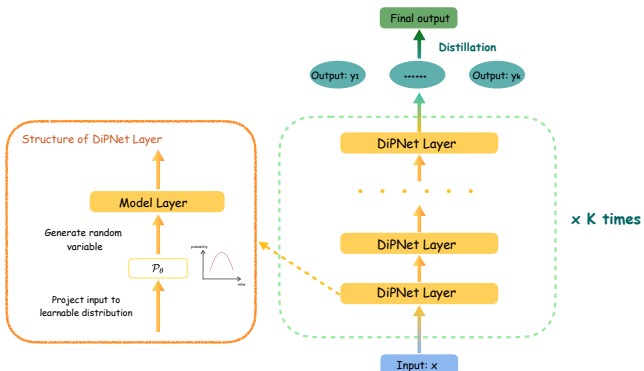

Figure 1: Pipeline of Distributional Input Projection Network.

output mappings lead to improved robustness and generalization. Smoothness mitigates the model's sensitivity to small input perturbations, thereby enhancing stability across data distributions. Nonetheless, directly enforcing smoothness in deep networks remains difficult due to their nonlinear and high-dimensional nature.

In this paper, we introduce Distributional Input Projection Networks (DIPNet)—a novel architectural framework that improves generalization by implicitly promoting smoothness. Rather than passing deterministic inputs through the network, DIPNet projects inputs into learnable distributions at each layer, enabling the model to reason over localized neighborhoods in the input space. This structured stochasticity acts as a form of regularization, smoothing the loss landscape and mitigating sensitivity to input variation. As shown in Figure 1, the proposed framework consistently improves generalization across a wide range of datasets and model architectures, with particularly notable gains under adversarial attacks and distribution shifts.

Moreover, DIPNet is broadly applicable and can be integrated into existing architectures with minimal overhead. Its distributional nature also draws inspiration from variational inference, which motivates our efficient training procedure with a stability-promoting penalty.

Our main contributions are as follows:

1. We introduce DIPNet, a novel architectural framework that enhances smoothness by projecting inputs into learnable distributions at each layer of the network.

2. We develop an efficient training method for DIPNet, inspired by variational inference, which also incorporates a stability penalty to promote robustness and regularization.

3. We provide theoretical guarantees showing that DIPNet reduces function smoothness measures and Lipschitz constants, offering insight into its generalization benefits.

4. We conduct extensive experiments across a diverse set of tasks and architectures—including ViTs, LLMs, MLPs, ResNets, adversarial robustness settings, out-of-distribution (OOD) generalization, and reasoning benchmarks—demonstrating consistent improvements in test-time performance. Such improvements are particularly significant under adversarial attacks or distribution shifts.

## 2 RELATED WORK

**Generalization performance.**    From the input covariates perspective, aligning with the conventional smoothness viewpoint, Johnson & Zhang (2023) recently suggested different indicators to measure generalization, i.e, "inconsistency" (Nakkiran & Bansal, 2020; Jiang et al., 2021; Kirsch & Gal, 2022) and "instability" (Bousquet & Elisseeff, 2002; Shalev-Shwartz et al., 2010), with the latter being related to the Lipschitz norm of the output models. While adversarial training (Madry et al., 2017; Cohen et al., 2019; Salman et al., 2019; Lecuyer et al., 2019; Gowal et al., 2020; Wang et al., 2021; Zou et al., 2021) can control the Lipschitz norm, it often involves a trade-off between

generalization performance and adversarial robustness (Lipschitz) (Tsipras et al., 2018; Zhang et al., 2019; Javanmard et al., 2020; Donhauser et al., 2021; Dobriban et al., 2023; Hao & Zhang, 2024). In contrast, our proposed method enhances generalization performance by projecting the input into a learnable distribution, thereby avoiding such trade-off typically observed with other approaches.

**Smoothing methods.** Prior work has explored enhancing smoothness through perturbation-based sampling in various contexts. For example, Shen & Meinshausen (2023) proposed Engression, a method that learns distributions via pre-additive perturbations. Similarly, diffusion models (Ho et al., 2020; Song et al., 2020; Dhariwal & Nichol, 2021; Saharia et al., 2022; Rombach et al., 2022) can be interpreted as techniques for modeling sample distributions from Gaussian noise. Another related line of research focuses on improving model robustness. In particular, certified robustness methods (Cohen et al., 2019; Salman et al., 2019; Lecuyer et al., 2019; Yang et al., 2020) aim to ensure stability under perturbations by injecting noise from specific distributions during training. Gaussian noise injection has also been widely used in data augmentation (Moreno-Barea et al., 2018) to enhance generalization by artificially expanding the training set. While our method also seeks to promote smoothness, it differs in two key ways: (i) Rather than treating noise injection as a training-only technique, we incorporate distributional projection directly into the model architecture, applying it consistently during both training and inference. This architectural integration enables theoretical guarantees for improved generalization. (ii) Instead of limiting perturbations to the input layer, we project the input into a learnable distribution at every layer of the network. This layerwise control provides enhanced stability and helps mitigate issues such as gradient explosion during training.

## 3 PROBLEM FORMULATION AND VARIATIONAL INFERENCE

In this section, we formally define the problem. Our goal is to learn a model parameterized by $\theta$ that minimizes the negative log-likelihood loss for prediction tasks:

$$\mathcal{L}(\theta) = -\mathbb{E}_x \ln \mathbb{P}(y|x, \theta).$$

**Derivation of distributional input.** Inspired by Shen & Meinshausen (2023), on each layer of the model, we consider to project input $x \in \mathbb{R}^p$ into a distribution $\mathcal{N}(x, \Sigma)$, where $\Sigma$ is the learnable variance, to take a better prediction. With a training set containing $n$ samples $\{x_i, y_i\}_{i=1}^n$, we can assume there exists an unobserved variable $\eta \sim \mathcal{N}(0, \gamma I_p)$, such that

$$\mathbb{P}(y|x, \eta, \theta) = \mathbb{P}(y|x + \eta, \theta). \tag{1}$$

Following the standard variational-inference derivation, we obtain the bound for $\mathcal{L}(\theta)$ as:

$$\mathcal{L}(\theta) = -\sum_{i=1}^n \ln \mathbb{P}(y_i|x_i, \theta) = -\sum_{i=1}^n \ln \mathbb{E}_\eta \mathbb{P}(y_i, \eta|x_i, \theta) = -\sum_{i=1}^n \ln \mathbb{E}_\eta \left( \mathbb{P}(y_i|x_i + \eta, \theta) \mathbb{P}(\eta) \right)$$

$$\leq -\sum_{i=1}^n \mathbb{E}_{\eta \sim q(\eta)} \left[ \ln \mathbb{P}(y_i|x_i + \eta, \theta) + \ln \mathbb{P}(\eta) - \ln q(\eta) \right],$$

for any distribution $q(\eta)$, where the third equality is from Eq. (1), and the last inequality is from ELBO lower bound[1]. For simplicity, we constrain $q(\eta)$ to be a Gaussian distribution:

$$q(\eta) := \mathcal{N}(0, \Sigma), \quad \text{where} \quad \Sigma = \text{diag}\{\lambda_1, \ldots, \lambda_p\},$$

then the lower bound should be:

$$-\sum_{i=1}^n \mathbb{E}_{\eta \sim \mathcal{N}(0, \Sigma)} \left[ \ln \mathbb{P}(y_i|x_i + \eta, \theta) - \frac{p \ln \gamma}{2} + \frac{\ln |\Sigma|}{2} - \frac{\eta^T \eta}{2\gamma} + \frac{\eta^T \Sigma^{-1} \eta}{2} \right]$$

$$= -\sum_{i=1}^n \mathbb{E}_{\eta \sim \mathcal{N}(0, \Sigma)} \left[ \ln \mathbb{P}(y_i|x_i + \eta, \theta) - \frac{p \ln \gamma}{2} + \frac{\sum_{j=1}^p \ln \lambda_j}{2} - \frac{\sum_{j=1}^p \lambda_j}{2\gamma} + \frac{p}{2} \right].$$

---

[1]Here is an upper bound as we consider the negative log-likelihood function.

Based on this formulation, to achieve accurate predictions, we minimize the following loss function:

$$\min_{\theta, \Sigma} \left\{ -\sum_{i=1}^{n} \mathbb{E}_{\eta \sim \mathcal{N}(0, \Sigma)} \ln \mathbb{P}(y_i | x_i + \eta, \theta) + \alpha \sum_{j=1}^{p} \lambda_j - \beta \sum_{j=1}^{p} \ln \lambda_j \right\}, \tag{2}$$

with uning parameter $\alpha, \beta > 0$.

**Remark 1.** *The term $\alpha \sum_{j=1}^{p} \lambda_j - \beta \sum_{j=1}^{p} \ln \lambda_j$ can be regarded as a special penalty term to prevent the corresponding parameters $\{\lambda_1, \ldots, \lambda_p\}$ shrinking to zero during training process.*

## 4 DIPNet: Distributional Input Projection Network

Building on the framework above, we propose a new algorithm—Distributional Input Projection Network (DIPNet), which projects the input into a learnable distribution at each layer of the neural network. An overview of the DIPNet pipeline is presented in Figure 1. We describe the key components and implementation details in this section.

### 4.1 Learning Distributional Input Layerwise to Minimize Loss Function

Motivated by enhancing generalization performance, DIPNet can switch the standard multi-layer neural network into a distributional input projection framework. To be specific, on each layer of the model, denoting the output of the previous layer as $v$, DIPNet performs the following steps:

1. Project $v$ into a learnable Gaussian distribution $\mathcal{N}(v, \Sigma)$;
2. Sample a particle $u$ from $\mathcal{N}(v, \Sigma)$;
3. Use $u$ as the input to the current layer and compute its output.

Since each layer introduces stochasticity through distributional projection, the final output becomes inherently random. To ensure stable and reliable predictions, this input-output procedure is repeated $k$ times. The final output is then obtained by taking average over all $k$ sampled trajectories.

**Adding a stability penalty.** While DIPNet can enforce smoothness by distributional projection, it induces instability on model output $f(x, \eta_1, \ldots, \eta_L, \theta)$ ($L$ is the number of model layers). To address this issue, we propose to penalizing the variance of the model output. To be specific, based on Eq. (2), we now formulate the loss function as:

$$\min_{\theta, \{\Sigma_l\}_{l=1}^{L}} \mathcal{L}_{\alpha, \beta, \lambda}(\theta, \Sigma_1, \ldots, \Sigma_L) := \min_{\theta, \{\Sigma_l\}_{l=1}^{L}} \left\{ -\sum_{i=1}^{n} \mathbb{E}_{\eta_1, \ldots, \eta_L} \ln \mathbb{P}(y_i | x_i, \eta_1, \ldots, \eta_L, \theta) \right.$$

$$\left. + \alpha \sum_{l=1}^{L} \sum_{j=1}^{p_l} \lambda_j^l - \beta \sum_{l=1}^{L} \sum_{j=1}^{p_l} \ln \lambda_j^l + \lambda \underbrace{\sum_{i=1}^{n} \mathbb{V}_{\eta_1, \ldots, \eta_L} \left[ f(x_i, \eta_1, \ldots, \eta_L, \theta) \right]}_{\text{stability penalty}} \right\}, \tag{3}$$

where $\lambda$ is a regularization parameter, and for each $l \in [L]$, we have $\eta_l \sim \mathcal{N}(0, \Sigma_l)$, and $\Sigma_l = \text{diag}\{\lambda_1^l, \ldots, \lambda_{p_l}^l\}$. With a proper choice of $\lambda$, adding such penalty in training process can avoid extremely instable model output, thereby benefits generalization (Johnson & Zhang, 2023).

**Unbiased estimation on loss function.** Before introducing the training algorithm formally, we first formulate the true loss function we use in practice:

$$-\frac{1}{m} \sum_{i=1}^{n} \sum_{j=1}^{m} \ln \mathbb{P}(y_i | x_i, \{\eta_{l,i,j}\}_{l=1}^{L}, \theta) + \alpha \sum_{l=1}^{L} \sum_{j=1}^{p_l} \lambda_j^l - \beta \sum_{l=1}^{L} \sum_{j=1}^{p_l} \ln \lambda_j^l$$

$$+ \frac{\lambda}{m(m-1)} \sum_{i=1}^{n} \sum_{1 \le j_1 < j_2 \le m} \| f(x_i, \{\eta_{l,i,j_1}\}_{l=1}^{L}, \theta) - f(x_i, \{\eta_{l,i,j_2}\}_{l=1}^{L}, \theta) \|_2^2, \tag{4}$$

where $\{\eta_{l,i,j}\}$ are i.i.d. sampled from $\mathcal{N}(0, \Sigma_l)$, and $m$ is the number of sample times. This leads to the *unbiased estimator* for Eq. (3). We now formally introduce the Distributional Input Projection Network (DIPNet) algorithm, which is summarized in Algorithm 1.

---

**Algorithm 1** Distributional Input Projection Network (DIPNet)

---

**Input:** Initial parameter $\{\theta^0, \Sigma_1^0, \dots, \Sigma_L^0\}$, training samples $\{x_i, y_i\}_{i=1}^n$, forecasting input $x'$, output model $f$, layer number $L$, repeating time in training $m$, repeating time in prediction $k$, epochs $T$, regularization $\{\alpha, \beta, \lambda\}$ and step size $\xi$.
2: ▷ **Training process:**
   **for** $t = 1, \dots, T$ **do**
4:    For each sample $\{x_i, y_i\}$, repeat Algorithm 2 $m$ times, to receive its corresponding output $\{f_j^t(x_i)\}_{j=1}^m$.
     Update parameter $\{\theta, \Sigma_1, \dots, \Sigma_L\}$ based on Eq (4) via gradient descent.
6: **end for**
   ▷ **Prediction process:**
8: Perform Algorithm 3 with input $x'$, then receive its corresponding output $f(x')$.
   **Output:** Final prediction $f(x')$.

---

**Model distillation in DIPNet.** During prediction, repeated sampling in Algorithm 2 incurs high time costs. To improve efficiency while maintaining accuracy, we adopt model distillation as described in Algorithm 3.

**Comparing with other methods.** Injecting Gaussian noise is a widely used technique in data augmentation (Moreno-Barea et al., 2018) and smoothing training (Cohen et al., 2019) to improve performance. In contrast, our proposed method goes beyond traditional augmentation in two key ways: (i) Rather than treating noise injection as a training-only strategy, we incorporate distributional projection directly into the model architecture, applying it consistently during both training and inference. This architectural integration enables theoretical guarantees for improved generalization. (ii) Instead of injecting noise at the input level only, we project the input into a learnable distribution at each layer, providing layerwise control over perturbations. This design offers better stability and helps prevent issues such as gradient explosion during training.

## 4.2 THEORETICAL RESULTS

We provide some theoretical guarantees on why our approaches improve the Lipschitz and smoothness of the original model. Our analysis is focusing on function[2] $h(\cdot, \theta) : \mathbb{R}^p \to \mathbb{R}$. Denoting $\eta \sim \mathcal{P}$, the distributional input projection function is denoted as

$$g_{\mathcal{P}}(x, \theta) := \int h(x + \eta, \theta)\mu_{\mathcal{P}}(\eta)d\eta,$$

where $\mu_{\mathcal{P}}$ is the probability density function of $\mathcal{P}$.

**Lipschitz norm.** Johnson & Zhang (2023) proposed that there are two terms, i.e., "instability" term and "inconsistency" term, which are strongly predictive of the generalization gap (the difference between the performance on the training data and unseen data). Following Theorem 1 in Johnson & Zhang (2023), the "instability" term is related to the function Lipschitz. The following theorems demonstrate that our distributional input projection methods can reduce the function's Lipschitz norm compared to the original function, contributing to improved generalization (The proofs are in Appendix B.1 and B.2).

**Theorem 1** (Improvement under bounded condition). *Assume that $\|h(x, \theta)\|_\infty < +\infty$, there will be*

$$\max_{x \in \mathbb{R}^p} \|\nabla_x g_{\mathcal{P}}(x, \theta)\|_2 \leq \|h(x, \theta)\|_\infty \|\nabla \mu_{\mathcal{P}}\|_{\mathcal{L}_2},$$

*where we denote $\|\nabla \mu_{\mathcal{P}}\|_{\mathcal{L}_2} := \int \|\nabla_t \mu_{\mathcal{P}}(t)\|_2 dt.$*

---

[2]For simplicity, here we only consider one-dim output, and the analysis for multiple dimension output function is similar.

Theorem 1 shows that even if the original function $h(\cdot)$ is non-Lipschitz, our method can still guarantee a Lipschitz function $g_{\mathcal{P}}(\cdot)$ under the mild condition that the output space of $h(\cdot)$ is bounded. Furthermore, Theorem 2 demonstrates that even when $h(\cdot)$ is already Lipschitz, DIPNeT can reduce its Lipschitz norm, thereby improving generalization:

**Theorem 2** (Improvement under Lipschitz condition). *Denote*

$$b := \max_x \|\nabla_x h(x, \theta)\|_2 < +\infty, \quad \mathcal{B}(c) := \{x \in \mathrm{R}^p | \|\nabla_x h(x, \theta)\|_2 > c \cdot b\},$$

*for any $0 < c < 1$. If there exists some constant $c$ such that $\mu(\mathcal{B}(c)) < +\infty$. We will obtain that*

$$\inf_{\mathcal{P}} \left\{ \max_{x \in \mathbb{R}^p} \|\nabla_x g_{\mathcal{P}}(x + \eta, \theta)\|_2 \right\} \leq c \cdot b.$$

**Smoothness.** It has been widely observed that lower function smoothness is often associated with better generalization performance in neural networks. Our proposed methods can also enforce the smoothness condition using a simple technique, as demonstrated in Theorem 3:

**Theorem 3** (Improvement under smoothness condition). *Denote*

$$s := \max_x \|\nabla_x^2 h(x, \theta)\|_2 < +\infty, \quad \mathcal{S}(c) := \{x \in \mathrm{R}^p | \|\nabla_x^2 h(x, \theta)\|_2 > c \cdot s\},$$

*for any $0 < c < 1$. If there exist some constant $c$ such that $\mu(\mathcal{S}(c)) < +\infty$. We will obtain that*

$$\inf_{\mathcal{P}} \left\{ \max_{x \in \mathbb{R}^p} \|\nabla_x^2 g_{\mathcal{P}}(x + \eta, \theta)\|_2 \right\} \leq c \cdot s.$$

The proof is in Appendix B.3. The results indicate that DIPNet can reduce the smoothness of models. In the sequel, we will show how this leads to better generalization performance.

---

**Algorithm 2** DIPNet: Practical Implementation

---

    **Input:** Model parameter $\{\theta, \Sigma_1, \ldots, \Sigma_L\}$, input $x$, output model $f$, layer number $L$.
2:  Initialize the input as $v_0$.
    **for** $l = 1, \ldots, L$ **do**
4:      Sample a particle $u_l$ from $\mathcal{N}(v_{l-1}, \Sigma_l)$.
       Take $u_l$ as the input of Layer-$l$.
6:      Receive the output on Layer-$l$, and denote it as $v_l$.
    **end for**
8:  **Output:** $v_L$.

---

**Algorithm 3** DIPNet: Model Distillation Prediction

---

    **Input:** Model parameter $\theta$, input $x$, output model $f$, layer number $L$.
2:  Initialize the input as $v_0$.
    **for** $l = 1, \ldots, L$ **do**
4:      Take $v_{l-1}$ as the input of Layer-$l$.
       Receive the output on Layer-$l$, and denote it as $v_l$.
6:  **end for**
    **Output:** $v_L$.

---

## 5 EXPERIMENT

We evaluate our proposed method against standard training as well as several baselines, including generalization methods—Sharpness-Aware Minimization (SAM) (Foret et al., 2020) and Randomized Smoothing (RS) (Cohen et al., 2019); data augmentation techniques—Mixup (Zhang et al., 2018), CutMix (Yun et al., 2019), and AugMix (Hendrycks et al., 2020); and masking-based regularization methods—Cutout (DeVries & Taylor, 2017) and Cutoff (Shen et al., 2020). Additional experiment explorations and additional ablation studies are in Appendix D.

## 5.1 Exploration on Vision Transformers under Adversarial Attacks

Compared to general non-perturbation methods, DIPNet achieves higher accuracy under adversarial attacks in training process, indicating their robustness. Here we consider two types of training-time attacks:

- **Randomized Gaussian noise**: we sample $\eta \sim \mathcal{N}(0, \sigma^2 I)$ for some $\sigma > 0$, and attack the input via $x \to x + \eta$ to simulate natural distribution shifts during training.

- **Fast Gradient Sign Method (FGSM) adversarial noise** (Goodfellow et al., 2014): we attack examples by $x \to x + \epsilon \cdot \text{sgn}\left(\nabla_x \mathcal{L}(\theta, x, y)\right)$. This single-step attack efficiently approximates the worst-case attacks within an $\ell_\infty$-ball of radius $\epsilon$.

By injecting either type of attack into the training inputs, we can evaluate adversarial robustness by showing the accuracy on clean test data. We evaluate our method in Vision Transformers (ViTs) (Dosovitskiy et al., 2020), a popular architecture in computer vision known for its strong performance across classification, detection, and segmentation tasks.

**Setup.** We train three ViT backbones—ViT-Tiny (5.5M parameters), ViT-Small (21.7M parameters), and ViT-Base (85.8M parameters)—on the CIFAR-100 dataset (Krizhevsky, 2009), an image classification dataset which contains 100 distinct classes of small-scale images. Each model is started from a checkpoint pretrained on ImageNet-21k and fine-tuned on ImageNet-1k, using a patch size of 16 and an input resolution of $224 \times 224$. During training, we inject either additive Gaussian noise ($\sigma = 0.2$) or FGSM adversarial attacks ($\epsilon = 0.2$) into the training inputs. At evaluation, we report accuracy on clean test sets. Additional implementation details are provided in Appendix D.4.

**Results.** Table 1 reports the performance of ViT models under three training-time attack settings.

Under the clean setting (no attack), our proposed method (DIPNet) consistently outperforms the Standard baseline across all ViT backbones, demonstrating improved generalization capabilities. In contrast, RS, which injects Gaussian perturbations randomly at the input layer, and AugMix, which relies on strong data augmentations, both significantly hurt clean accuracy—particularly evident on ViT-Tiny (65.45% and 62.34% vs. 84.71% for Standard). Under Gaussian noise, most baselines perform similarly to the Standard with only marginal changes, whereas DIPNet delivers clear robustness gains on ViT-Tiny and ViT-Small and remains comparable to the baseline for ViT-Base. Further details on the ViT-Base Gaussian scenario are in Appendix D.7.3. Under FGSM, DIPNet improves over Standard and surpasses other baselines on ViT-Tiny and ViT-Small, remaining competitive on ViT-Base. See Appendix D.4 for training and validation curves.

Overall, DIPNet both maintains high accuracy under the None-attack setting and provides strong robustness against both Gaussian and FGSM attacks, consistently achieving the best average performance across all three ViT backbones.

## 5.2 Exploration on LLM Reasoning

We extend our method to large-scale language models (LLMs), which often contain billions of parameters and now power a wide range of applications—from machine translation and summarization to dialogue systems and deep thinking. In particular, mathematical reasoning tasks like GSM8K (Cobbe et al., 2021) have recently attracted significant attention for assessing an LLM's ability to perform precise arithmetic and logical reasoning. We hypothesize that our method will help LLMs form smoother latent representations, improving both generalization to new problems and robustness against prompt variations.

**Setup.** We use the official OpenAI GSM8K dataset, which consists of 1,319 grade-school math word problems requiring several reasoning steps and an exact numerical answer, replacing the original "#### <answer>" marker with "The answer is <answer>". To test our method across diverse architectures, we select six popular open-source models: Qwen2.5-3B and Qwen2.5-7B (Yang et al., 2024), Llama-3.2-3B and Llama-3.1-3B (Dubey et al., 2024), and Gemma-3-4B and Gemma-3-12B (Farabet & Warkentin, 2025). Due to computational constraints, we fine-tune each model with LoRA for a single epoch—using a learning rate of $5 \times 10^{-4}$, Then we evaluate on the

Table 1: Test accuracy (%) on CIFAR-100 under different training-time attacks.

| | Method | None | Gaussian | FGSM | 3 Average |
|---|---|---|---|---|---|
| | Standard | 84.71 | 46.31 | 20.89 | 50.64 |
| | Sharpness-Aware Minimization (SAM) | 84.46 | 46.04 | 22.63 | 51.04 |
| | Randomized Smoothing (RS) | 65.45 | 47.04 | 23.22 | 45.24 |
| ViT-Tiny | Cutout | 85.39 | 43.47 | 21.64 | 50.17 |
| | Mixup | **85.54** | 46.34 | 23.55 | 51.81 |
| | CutMix | 85.36 | 42.77 | 20.33 | 49.49 |
| | AugMix | 62.34 | 52.11 | **33.84** | 49.43 |
| | DIPNet | 85.45 | **52.22** | 28.18 | **55.28** |
| | Standard | 89.59 | 75.65 | 65.64 | 76.96 |
| | Sharpness-Aware Minimization (SAM) | 89.72 | 74.22 | 66.86 | 76.93 |
| | Randomized Smoothing (RS) | 84.84 | 74.31 | 65.55 | 74.90 |
| ViT-Small | Cutout | 90.17 | 74.54 | 67.15 | 77.29 |
| | Mixup | **90.64** | 70.74 | 63.45 | 74.94 |
| | CutMix | 90.42 | 74.36 | 63.61 | 76.13 |
| | AugMix | 87.14 | **78.71** | 69.17 | 78.34 |
| | DIPNet | 90.14 | 78.07 | **69.65** | **79.29** |
| | Standard | 92.46 | 69.13 | 70.75 | 77.45 |
| | Sharpness-Aware Minimization (SAM) | 92.68 | 69.28 | 70.34 | 77.43 |
| | Randomized Smoothing (RS) | 88.32 | 68.74 | **77.30** | 78.12 |
| ViT-Base | Cutout | **93.16** | 65.39 | 76.97 | 78.51 |
| | Mixup | 92.53 | **71.23** | 63.04 | 75.60 |
| | CutMix | 92.75 | 62.17 | 71.75 | 75.56 |
| | AugMix | 89.12 | 69.60 | 74.40 | 77.71 |
| | DIPNet | 92.87 | 69.23 | 74.20 | **78.75** |

*Notes.* **Bold** indicates the best performance, and underlined indicates the second best.

Table 2: Accuracy (%) on GSM8K.

| Method | Qwen2.5-3B | Llama-3.2-3B | Gemma-3-4B | Qwen2.5-7B | Llama-3.1-8B | Gemma-3-12B |
|---|---|---|---|---|---|---|
| SFT | 70.96 | 32.15 | 44.05 | 78.92 | 52.99 | 72.78 |
| RS | 72.10 | 32.68 | 44.43 | 78.84 | 54.36 | 73.31 |
| SAM | 68.84 | 32.83 | 45.11 | 78.09 | 54.13 | 72.10 |
| Cutoff | 69.75 | 31.92 | 44.28 | 78.77 | 54.51 | 72.93 |
| DIPNet | **72.17** | **33.06** | **46.32** | **79.61** | **54.74** | **74.22** |

*Notes.* **Bold** indicates the best performance, and underlined indicates the second best.

GSM8K test set using zero-shot CoT prompting, and use Math-Verify to validate the generation [3]. Additional implementation details are provided in Appendix D.5.

**Results.** Table 2 reports the accuracy of DIPNet and several baselines. Our method consistently outperforms other baselines on all models, yielding nontrivial gains. Across different model types, we observe larger improvements for the Gemma-3 series. The results demonstrate that our method successfully improves LLMs' performance by enhancing the smoothness during training. And our method is still promising for models with parameters > 10 Billion.

### 5.3 ABLATION ON EFFICIENT INFERENCE

To improve inference efficiency, we consider model distillation as an alternative to repeated sampling. Specifically, we compare distilled inference against the original method, which performs the inference process in Algorithm 2 $k$ times and averages the outputs as the final prediction. Results on ViT-Tiny under Gaussian attack are reported in Figure 2a, and results on the Llama-3.2-3B model are shown in Figure 2b. For ViT models, distillation achieves performance comparable to multi-sample averaging, but the latter is far less efficient, especially for large models. For example, on ViT-B, sampling 50 times achieves similar accuracy but requires about $80\times$ more inference time than distillation. See

---

[3]https://github.com/huggingface/Math-Verify

Appendix D.6.1 for detailed experimental results. In Figure 2b, increasing the number of samples $k$ generally improves performance; however, even with 50 samples, the accuracy lags behind that of distillation while incurring prohibitive inference costs. These results suggest that distillation provides a more efficient alternative, delivering competitive or superior accuracy while significantly reducing inference time.

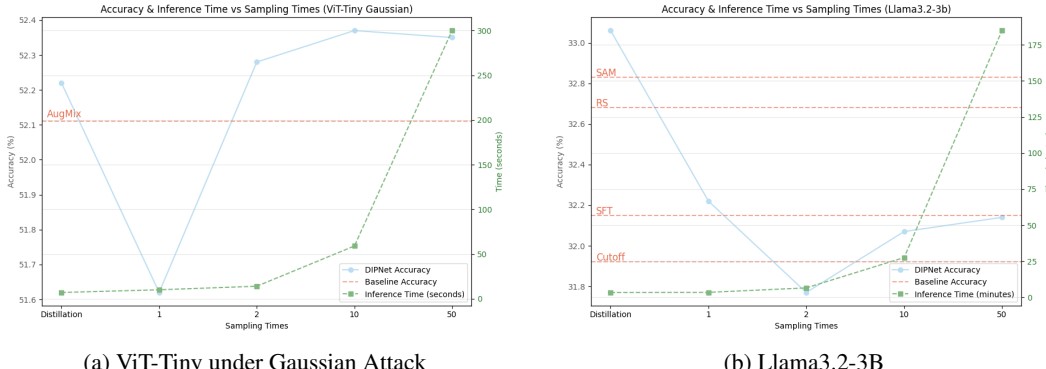

(a) ViT-Tiny under Gaussian Attack        (b) Llama3.2-3B

Figure 2: Accuracy & Inference Time vs Sampling times $k$ and time cost using (a) ViT-Tiny under Gaussian attack and (b) Llama3.2-3B, compared with Distillation and baselines.

### 5.4 ABLATION ON HYPERPARAMETER TUNING

We study the sensitivity of DIPNet to the scalar hyperparameters—$\alpha$, $\beta$, and $\lambda$—which balance the distributional projection penalty and the stability term in the training objective, using the grid in Table 3. For ViT-Tiny under Gaussian attack, varying $(\alpha, \beta) \in \{0.05, 0.10, 0.50\} \times \{0.10, 0.20, 0.50\}$ yields only minor accuracy changes when $\lambda$ is fixed. The best results are generally obtained around moderate values, indicating that DIPNet is not overly sensitive to small perturbations of $(\alpha, \beta)$ and that a mid-range setting provides a stable and reliable default. In practice, we recommend starting with $(\alpha, \beta) = (0.1, 0.2)$ and adjusting $\beta$ relative to $\alpha$ only if stronger or weaker regularization is desired.

By contrast, $\lambda$ exhibits a stronger influence. Table 3 shows $\lambda = 0$ consistently achieves the best accuracy across different $(\alpha, \beta)$ combinations. However, when training ViTs from scratch (see details in Appendix D.6.2), introducing a small $\lambda$ (e.g., $\lambda = 0.05$) can improve performance over the $\lambda = 0$ baseline, suggesting that the penalty acts as an effective regularizer during early training.

Table 3: Detailed experimental results for ViT-Tiny under Gaussian attack.

| $\lambda \setminus (\alpha, \beta)$ | (0.05,0.1) | (0.05,0.2) | (0.05,0.5) | (0.1,0.1) | (0.1,0.2) | (0.1,0.5) | (0.5,0.1) | (0.5,0.2) | (0.5,0.5) |
|---|---|---|---|---|---|---|---|---|---|
| 0 | 51.99 | 52.05 | 51.86 | 52.13 | 52.10 | 52.16 | 52.22 | 52.09 | 52.21 |
| 0.001 | 51.12 | 51.10 | 51.28 | 51.33 | 51.40 | 51.23 | 51.11 | 51.22 | 51.17 |
| 0.01 | 46.39 | 46.32 | 46.33 | 46.33 | 46.34 | 46.33 | 46.36 | 46.34 | 46.33 |

## 6 CONCLUSION

In this work, we proposed Distributional Input Projection Networks (DIPNet), a novel architectural framework designed to enhance generalization by projecting inputs into learnable distributions at each layer. This approach implicitly enforces smoothness and reduces the Lipschitz constant of the network, supported by both theoretical analysis and extensive empirical validation. Across a wide range of models and tasks, DIPNet consistently improves test performance in standard, adversarial, and out-of-distribution settings. Our results demonstrate that DIPNet offers a broadly applicable and effective strategy for improving generalization in overparameterized deep networks. We consider extending this framework to more complex reasoning tasks and reinforcement learning as a promising direction for future work.

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

## A  ADDITIONAL RELATED WORK

**Influential factors on generalization performance.**  In real-world scenarios, over-parameterized neural networks often find solutions that perform optimally or near-optimally on training data, but these solutions do not always generalize well to test data. Given the significant challenges associated with generalization performance in deep neural networks, many studies have aimed to identify the key factors influencing generalization. From the parameter perspective, a considerable body of research has focused on the relationship between function sharpness and generalization (Hochreiter & Schmidhuber, 1997; Keskar et al., 2016; Izmailov et al., 2018; Jiang et al., 2019; Foret et al., 2020). For instance, Foret et al. (2020) intrtoduced a method named *sharpness-aware-minimization* (SAM), which could reduce the sharpness of output model and subsequently improves generalization. However, even if the solution of SAM always aligns with flat loss landscape, it may not be optimal (Yue et al., 2023). From the input covariates perspective, aligning with the conventional smoothness viewpoint, Johnson & Zhang (2023) recently suggested different indicators to measure generalization, i.e, "inconsistency" (Nakkiran & Bansal, 2020; Jiang et al., 2021; Kirsch & Gal, 2022) and "instability" (Bousquet & Elisseeff, 2002; Shalev-Shwartz et al., 2010), with the latter being related to the Lipschitz norm of the output models. While adversarial training (Madry et al., 2017; Cohen et al., 2019; Salman et al., 2019; Lecuyer et al., 2019; Gowal et al., 2020; Wang et al., 2021; Zou et al., 2021) can control the Lipschitz norm, it often involves a trade-off between generalization performance and adversarial robustness (Lipschitz) (Tsipras et al., 2018; Zhang et al., 2019; Javanmard et al., 2020; Donhauser et al., 2021; Dobriban et al., 2023; Hao & Zhang, 2024). In contrast, our proposed method enhances generalization performance by projecting the input into a learnable distribution, thereby avoiding such trade-off typically observed with other approaches.

## B  THEORETICAL ANALYSIS

### B.1  PROOF OF THEOREM 1

Consider the expression of function $g$, for any $x \in \mathbb{R}^p$, we have

$$g_\mathcal{P}(x, \theta) = \int h(x + \eta, \theta) \mu_\mathcal{P}(\eta) d\eta$$

$$= \int h(t, \theta) \mu_\mathcal{P}(t - x) dt.$$

So we could obtain the gradient norm of $g(x, \theta)$ as

$$\|\nabla_x g_\mathcal{P}(x, \theta)\|_2 = \left\| \nabla_x \int h(t, \theta) \mu_\mathcal{P}(t - x) dt \right\|_2$$

$$= \left\| \int h(t, \theta) \nabla_x \mu_\mathcal{P}(t - x) dt \right\|_2$$

$$\leq \int \|h(t, \theta) \nabla_x \mu_\mathcal{P}(t - x)\|_2 dt$$

$$\leq \|h(x, \theta)\|_\infty \cdot \int \|\nabla_t \mu_\mathcal{P}(t)\|_2 dt.$$

### B.2  PROOF OF THEOREM 2

For any $0 < \epsilon < 1$, we can choose $\zeta > \epsilon^{-1}$, and denote $\mu(\mathcal{B}(c)) = C < +\infty$. Then we consider $\mathcal{P}$ as a uniform distribution on $\mathbb{R}^p$, which is supported on a set with measurement $\zeta C$, then we can obtain that

$$\|\mathbb{E}_\eta \nabla_x h(x + \eta, \theta)\|_2^2 \leq \mathbb{E}_\eta \|\nabla_x h(x + \eta, \theta)\|_2^2$$

$$= \mathbb{E}_\eta \left[ \|\nabla_x h(x + \eta, \theta)\|_2^2 | \mathbb{1}(x + \eta \in \mathcal{B}(c)) \right] \mathbb{P}(\mathcal{B}(c))$$

$$+ \mathbb{E}_\eta \left[ \|\nabla_x h(x + \eta, \theta)\|_2^2 | \mathbb{1}(x + \eta \notin \mathcal{B}(c)) \right] \mathbb{P}((\mathcal{B}(c))^c)$$

$$\leq \frac{1}{\zeta C} (C \cdot b + \zeta C \cdot cb) = cb + \frac{1}{\zeta} b < (c + \epsilon)b.$$

Sending $\epsilon \to 0$, we can finish the proof.

### B.3 Proof of Theorem 3

The proof is similar to the proof for Theorem 2. For any $0 < \epsilon < 1$, we can choose $\zeta > \epsilon^{-1}$, and denote $\mu(\mathcal{S}(c)) = C < +\infty$. Then we consider $\mathcal{P}$ as a uniform distribution on $\mathrm{R}^p$, which is supported on a set with measurement $\zeta C$, then we can obtain that

$$
\begin{aligned}
\|\mathbb{E}_\eta \nabla_x^2 h(x + \eta, \theta)\|_2^2 &\leq \mathbb{E}_\eta \|\nabla_x^2 h(x + \eta, \theta)\|_2^2 \\
&= \mathbb{E}_\eta \left[ \|\nabla_x^2 h(x + \eta, \theta)\|_2^2 | 1(x + \eta \in \mathcal{S}(c)) \right] \mathbb{P}(\mathcal{S}(c)) \\
&\quad + \mathbb{E}_\eta \left[ \|\nabla_x^2 h(x + \eta, \theta)\|_2^2 | 1(x + \eta \notin \mathcal{S}(c)) \right] \mathbb{P}((\mathcal{S}(c))^c) \\
&\leq \frac{1}{\zeta C} \left( C \cdot s + \zeta C \cdot cs \right) = cs + \frac{1}{\zeta} s < (c + \epsilon)s.
\end{aligned}
$$

Sending $\epsilon \to 0$, we can finish the proof.

## C Dataset Description

### C.1 Tabular Dataset for MLP

- **Air Quality.** (Vito, 2016) The Air Quality dataset comprises 15 features derived from hourly averaged responses of an array of 5 metal oxide chemical sensors embedded in an Air Quality Chemical Multisensor Device in an Italian city, collected from March 2004 to February 2005. In this study, we use 9 of these features, with "PT08.S3(NOx)" as the response variable and the remaining 8 features as input variables.

- **ETD.** (Zhou et al., 2021) The ETD dataset is related to the electronic data distribution hourly data recordings. It contains 8 features, including the date of the point, 6 different types of external power load features and the predictive value "oil temperature".

- **Sharing Bike.** (Fanaee-T & Gama, 2014) This dataset aims to understand the factors influencing the demand for shared bikes in the American market, with hourly data recorded from January 2011 to December 2012. For this analysis, we consider 4 features, i.e., weather, temperature, humidity, and feeling temperature, to predict the total count of rental bikes.

- **NASDAQ100.** (Qin et al., 2017) It contains stock prices of 81 corporations under NASDAQ 100 and the index value of NASDAQ 100. The frequency of the data collection is one-minute. Here we consider the stock prices of different corporations as input variables, and the index value of NASDAQ 100 as predictive variable.

Table 4: Overview of tabular dataset.

| Dataset | Air Quality | ETD | Sharing Bike | NASDAQ100 |
|---|---|---|---|---|
| input | vector | vector | vector | vector |
| task | reg. | reg. | reg. | reg. |
| # samples | 8991 | 17420 | 17379 | 40560 |

Table 5: Overview of OOD tabular dataset.

| Dataset | $VPower_s$ | $VPower_r$ | Weather |
|---|---|---|---|
| input | vector | vector | vector |
| task | reg. | reg. | reg. |
| # samples | 546543 | 554642 | 436733 |

### C.2 OOD Dataset

To assess robustness under distribution shift, we use the Shifts vessel power estimation dataset (Malinin et al., 2022) and the Shifts Weather Prediction dataset (Malinin et al., 2021) in OOD situations (see details in Table 5 and Figure 3).

- **Vessel Power Estimation**[4] The Shifts Vessel Power Prediction dataset contains 11 different features and one scalar target (energy utilization of cargo ships). And it contains two types of data: one is synthetic dataset ($VPower_s$), the other is real dataset ($VPower_r$).

- **Weather Prediction**[5] The Shifts Weather Prediction dataset (Malinin et al., 2021) provides a scalar regression task to forecast the air temperature. The dataset contains 127 features and a single regression target, spanning an entire year, i.e, from September 1st, 2018, to September 1st, 2019.

---

[4]Notice that this is a developing dataset and it only contains the training data and valid data recently. So here we use the ID valid dataset to take validation, and the OOD valid dataset is used for testing.

[5]More details can be found in `https://github.com/Shifts-Project/shifts`.

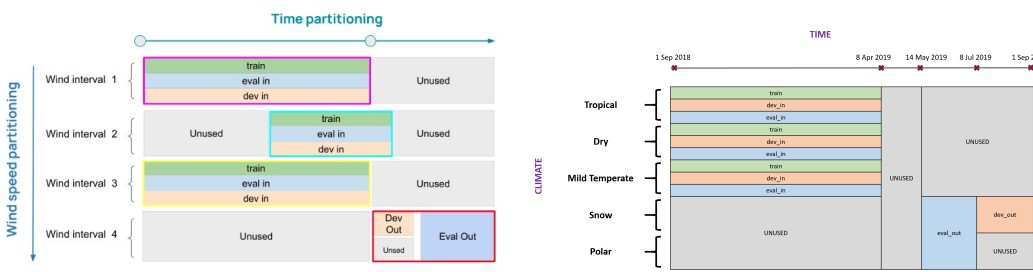

*Notes.* The figure on the left side is about the Shifts vessel power estimation dataset, and the figure on the right right is about the Shifts Weather Prediction dataset. "train", "dev" and "eval" refer to training data, validation data and test data respectively. As public access to canonical "test" of vessel power dataset is restricted, here we valid on "dev-in" and use "dev-out" to test the model performance.

Figure 3: OOD dataset descriptions

# D SUPPLEMENT EXPERIMENTS

## D.1 EXPLORATIONS ON MLP

Here we compare the generalization performance of our method and a standard non-perturbation output model by evaluating them on various datasets and different neural network architectures.

To measure the generalization performance of different methods, we take explorations on four tabular datasets and detailed description could be found in Appendix C.

Specifically, we test different settings, including various proportions of training and test data, the number of hidden layers, and the number of neurons in each layer. Using gradient descent on the $\ell_2$ loss, we summarize the results in Table 6, indicating that both the standard errors and the adversarial errors on DIPNet are consistently smaller than those for standard networks. This further verifies the benefits of our proposed method.

Table 6: Standard and Adversarial MSE on MLP.

| Dataset | Test proposition | Method | St(2+100) | Adv(2+100) | St(4+100) | Adv(4+100) | St(4+400) | Adv(4+400) |
|---|---|---|---|---|---|---|---|---|
| Air Quality | 0.3 | Standard | 0.0858 | 24.0112 | 0.0677 | 37.1026 | 0.0628 | 20.5767 |
| | | DIPNet | **0.0822** | **11.14** | **0.0576** | **20.9601** | **0.0526** | **20.5601** |
| | 0.5 | Standard | 0.1033 | 13.8045 | 0.0873 | 49.1966 | 0.0823 | 17.7776 |
| | | DIPNet | **0.0964** | **9.6509** | **0.0794** | **5.8675** | **0.0748** | **12.9039** |
| | 0.7 | Standard | 0.0984 | 16.6161 | 0.0938 | 24.4145 | 0.0910 | 14.5242 |
| | | DIPNet | **0.0901** | **4.8616** | **0.0844** | **6.6429** | **0.0853** | **10.1404** |
| ETD | 0.3 | Standard | 0.1845 | 4.7174 | 0.1693 | 5.1239 | 0.1635 | 4.5290 |
| | | DIPNet | **0.1787** | **3.6371** | **0.1604** | **3.7268** | **0.1578** | **2.9117** |
| | 0.5 | Standard | 0.1905 | 6.1011 | 0.1724 | 4.5385 | 0.1668 | 4.5335 |
| | | DIPNet | **0.1831** | **3.5267** | **0.1648** | **3.3667** | **0.1632** | **3.1108** |
| | 0.7 | Standard | 0.2023 | 5.0696 | 0.1913 | 6.2199 | 0.1867 | 5.1136 |
| | | DIPNet | **0.1947** | **4.3076** | **0.1776** | **2.9721** | **0.1757** | **2.9267** |
| Sharing Bike | 0.3 | Standard | 0.7221 | 2.5177 | 0.7208 | 2.0264 | 0.7216 | 2.1557 |
| | | DIPNet | **0.7188** | **2.0034** | **0.7186** | **1.8647** | **0.7182** | **1.8732** |
| | 0.5 | Standard | 0.7108 | 2.3387 | 0.7107 | 2.3030 | 0.7118 | 1.8314 |
| | | DIPNet | **0.7084** | **1.7799** | **0.7082** | **1.8780** | **0.7089** | **1.7272** |
| | 0.7 | Standard | 0.7075 | 1.7905 | 0.7057 | 1.8397 | 0.7068 | 1.7931 |
| | | DIPNet | **0.7050** | **1.7353** | **0.7043** | **1.7941** | **0.7056** | **1.7785** |
| NASDAQ100 | 0.3 | Standard | 3.576e-5 | 0.6066 | 3.538e-5 | **0.5447** | 2.779e-5 | 0.5303 |
| | | DIPNet | **2.757e-5** | **0.5824** | **2.511e-5** | 0.5628 | **2.158e-5** | **0.5195** |
| | 0.5 | Standard | 3.691e-5 | 0.5901 | 3.671e-5 | **0.5364** | 2.916e-5 | 0.5389 |
| | | DIPNet | **2.890e-5** | **0.5866** | **2.642e-5** | 0.5805 | **2.266e-5** | **0.5244** |
| | 0.7 | Standard | 6.146e-5 | 0.5813 | 8.085e-5 | 0.5506 | 3.612e-5 | 0.5455 |
| | | DIPNet | **3.122e-5** | **0.5812** | **2.886e-5** | **0.5428** | **2.532e-5** | **0.5336** |

*Notes.* "St" ,"Adv" refer to standard loss and adversarial loss respectively. "2+100", "4+100", "4+400" refer to the network architecture (e.g, "2+100" means the two layer network which each layer contains 100 neurons).

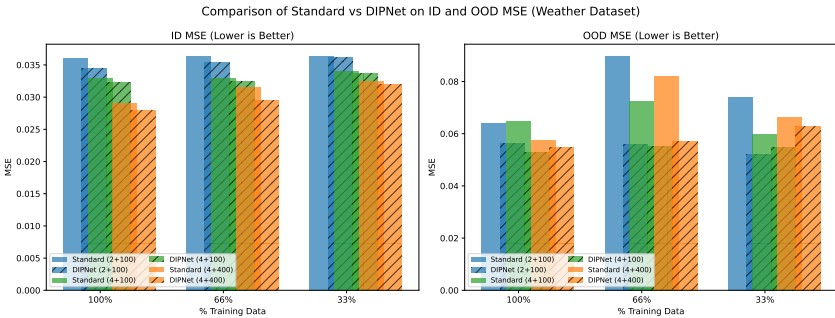

Figure 4: ID and OOD MSE on Weather Dataset.

*Notes.* Here we train on the ID dataset, validate on both ID and OOD datasets, and test the model performance on both ID and OOD datasets (with the test OOD environment differing from the validation OOD environment).

## D.2 OUT-OF-DISTRIBUTION (OOD) EVALUATION

We also explore the generalization performance on OOD tasks. We conduct experiments on tabular datasets using different neural network architectures, further validating the advantages of our proposed methods in both In-Distribution (ID) and Out-Of-Distribution (OOD) generalization performance.

**Setup.** To assess robustness under distribution shift, we use the Shifts vessel power estimation dataset (Malinin et al., 2022) and the Shifts Weather Prediction dataset (Malinin et al., 2021) in OOD situations. For each dataset, we train three MLP architectures—2 layers with 100 neurons per layer (2+100), 4 layers with 100 neurons per layer (4+100), and 4 layers with 400 neurons per layer (4+400)—using standard stochastic gradient descent on the MSE loss. We compare against the non-perturbed baseline (Standard). We vary the fraction of training data (100%, 66%, 33%) and evaluate the same three MLP architectures under each split.

**Results.** Tables 7, 8 and Figure 4 present both ID and OOD test MSEs. The results shows that comparing with standard non-perturbation method, DIPNet consistently improve ID performance, and enhance OOD generalization significantly.

Table 7: OOD MSE on Vessel Power dataset.

| % Train | | 100% | | | 66% | | | 33% | | |
|---|---|---|---|---|---|---|---|---|---|---|
| **Dataset** | **Method** | **2+100** | **4+100** | **4+400** | **2+100** | **4+100** | **4+400** | **2+100** | **4+100** | **4+400** |
| $Vpower_s$ | Standard | 0.0212 | 0.0231 | 0.0288 | 0.0401 | 0.0533 | 0.0601 | 0.0539 | 0.0498 | 0.0420 |
| | DIPNet | **0.0195** | **0.0206** | **0.0212** | **0.0232** | **0.0305** | **0.0298** | **0.0298** | **0.0318** | **0.0324** |
| $Vpower_r$ | Standard | 0.0400 | 0.0560 | 0.0895 | 0.0439 | 0.0401 | 0.0730 | 0.0944 | 0.0947 | 0.0859 |
| | DIPNet | **0.0363** | **0.0439** | **0.0555** | **0.0345** | **0.0383** | **0.0412** | **0.0581** | **0.0534** | **0.0593** |

*Notes.* "2+100", "4+100", "4+400" refer to the network architecture (e.g, "2+100" means the two layer network which each layer contains 100 neurons).

Table 8: ID and OOD MSE on Weather dataset.

| % Train | Method | ID(2+100) | ID(4+100) | ID(4+400) | OOD(2+100) | OOD(4+100) | OOD(4+400) |
|---|---|---|---|---|---|---|---|
| 100 % | Standard | 0.0360 | 0.0330 | 0.0291 | 0.0640 | 0.0647 | 0.0576 |
| | DIPNet | **0.0344** | **0.0322** | **0.0280** | **0.0562** | **0.0528** | **0.0547** |
| 66 % | Standard | 0.0364 | 0.0329 | 0.0315 | 0.0898 | 0.0723 | 0.0822 |
| | DIPNet | **0.0354** | **0.0324** | **0.0295** | **0.0558** | **0.0552** | **0.0569** |
| 33 % | Standard | 0.0364 | 0.0341 | 0.0325 | 0.0741 | 0.0599 | 0.0662 |
| | DIPNet | **0.0361** | **0.0337** | **0.0320** | **0.0519** | **0.0549** | **0.0627** |

*Notes.* Here we train on the ID dataset, validate on both ID and OOD datasets, and test the model performance on both ID and OOD datasets (with the test OOD environment differing from the validation OOD environment). "2+100", "4+100", "4+400" refer to the network architecture (e.g, "2+100" means the two layer network which each layer contains 100 neurons). "ID" means test data on ID environment , and "OOD" means test data on OOD environment.

### D.3 EXPLORATIONS ON RESNET

Furthermore, we extend our methods to ResNet architecture. To examine its performance compared to the standard non-perturbation ResNet, we conducted experiments on CIFAR10 and CIFAR100 using different ResNet structures. We then compared the classification accuracy (in percentage) on the test data for each method. The results, summarized in Table 9, demonstrate the benefits of DIPNet in improving generalization performance. These findings highlighting the extended potential of our proposed methods.

Table 9: Result classification accuracy (%) on ResNet.

| Method | res18-cifar10 | res34-cifar10 | res50-cifar10 | wideres-cifar10 | wideres-cifar100 |
|--------|--------------|--------------|--------------|----------------|-----------------|
| Standard | 90.96 | 93.52 | 93.37 | 96.14 | 80.77 |
| DIPNet | **91.72** | **93.56** | **93.89** | **96.53** | **81.73** |

### D.4 DETAILS FOR VIT EXPERIMENTS

All Vision Transformer experiments are conducted on a single NVIDIA A100-40G GPU. Checkpoints are loaded via the `timm` library. Our implementation is adapted from the publicly available repository[6], using default hyperparameters, enabling Apex O2 mixed-precision training (FP16).

For the two baselines—SAM and RS—we first perform a hyperparameter sweep on the ViT-Tiny model, then scale the optimal settings to larger backbones. Specifically, SAM's perturbation radius $\rho$ is searched over $\{0.01, 0.03, 0.05\}$, yielding $\rho = 0.05$; RS's noise level $\sigma$ is searched over $\{0.001, 0.005, 0.01, 0.05, 0.1\}$, yielding $\sigma = 0.01$. For the remaining four baselines, we follow the recommended settings in their papers on CIFAR-100. Specifically, Cutout randomly masks out a square region covering $50\%$ of the image area. Mixup and CutMix both sample mixing ratios from a Beta distribution with parameter $\alpha = 1.0$. For AugMix, we adopt the default configuration of severity $= 3$, width $= 3$, and depth uniformly sampled from $\{1, 3\}$.

Representative training and validation curves can be found in Figure 5, 6, 7 and 8.

### D.5 DETAILS FOR LLM EXPERIMENTS

All language-model experiments are conducted on a single NVIDIA A6000 GPU. We apply LoRA with rank 8, alpha 16, dropout 0.1, targeting the modules `["q_proj","v_proj"]`.

Training arguments include a batch size of 64, 1 epoch, weight decay of 0.01, warmup ratio of 0.03, and a fixed seed of 42. We search peak learning rates in $\{7 \times 10^{-5}, 1 \times 10^{-4}, 2 \times 10^{-4}, 3 \times 10^{-4}, 5 \times 10^{-4}\}$, which yields $5 \times 10^{-4}$ for all the DIPNet models.

Evaluation is performed on the GSM8K task with 0-shot CoT prompting.

For the two baselines—SAM and RS—we conduct hyperparameter sweeps and report the best-performing accuracy. In this case, SAM's $\rho$ is searched over $\{0.01, 0.02, 0.05\}$ and RS's $\sigma$ over $\{0.01, 0.02, 0.05\}$.

### D.6 DETAILED EXPERIMENTAL RESULTS

#### D.6.1 ABLATION ON EFFICIENT INFERENCE

Detailed experimental results can be found in Table 10 and 11.

#### D.6.2 ABLATION ON HYPERPARAMETER TUNING

Detailed experimental results can be found in Table 12.

| Model | Attack Type | Model Distillation | 1 | 2 | 10 | 50 |
|-------|-------------|--------------------|-----|-----|-----|-----|
| ViT-Tiny | None | 85.45 | 85.36 | 85.40 | 85.44 | 85.42 |
| | Gaussian | 52.22 | 51.62 | 52.28 | 52.37 | 52.35 |
| | FGSM | 28.18 | 27.19 | 27.21 | 27.44 | 27.45 |
| ViT-Small | None | 90.14 | 90.06 | 90.10 | 90.12 | 90.12 |
| | Gaussian | 78.07 | 78.21 | 78.33 | 78.31 | 78.36 |
| | FGSM | 69.65 | 69.17 | 69.21 | 69.25 | 69.17 |
| ViT-Base | None | 92.87 | 92.84 | 92.87 | 92.88 | 92.86 |
| | Gaussian | 69.23 | 69.31 | 69.26 | 69.24 | 69.19 |
| | FGSM | 74.20 | 74.17 | 74.16 | 74.18 | 74.18 |

Table 10: Accuracy (%) under different sampling times $k$.

| Model | Model Distillation | 1 | 2 | 10 | 50 |
|-------|--------------------|-----|-----|------|-------|
| ViT-Tiny | 7s | 10s | 14s | 59s | 5m |
| ViT-Small | 10s | 18s | 32s | 2m33s | 12m37s |
| ViT-Base | 27s | 52s | 1m42s | 8m19s | 41m37s |

Table 11: Inference time under different sampling times $k$.

## D.7 ADDITIONAL ABLATION STUDY

### D.7.1 ABLATION STUDY ON LAYERWISE DISTRIBUTIONAL INPUT PROJECTION

To assess how the depth of distributional input projections affects robustness, we compare four configurations of Layerwise DIPNet: taking the distributional input projection only at the first layer ("1-Layer"), at the first two layers ("2-Layer"), at every layer ("Full-Layer"), and the original model without any distributional projection ("Standard").

As Table 13a shows, taking the distributional input projection at only a single layer ("1-Layer") yields little to no generalization benefit—and can even lead to a slight drop in accuracy. In contrast, the "2-Layer" configuration delivers a clear generalization boost, outperforming Standard across all ViT scales, and on both ViT-Tiny and ViT-Base it even edges out the "Full-Layer" configuration—most notably pushing ViT-Base to 71.78 %. Overall, taking the distributional input projection at multiple depths delivers substantial gains, although the optimal number of layers can vary by model scale.

### D.7.2 ABLATION STUDY ON LEARNABLE DISTRIBUTIONAL INPUT PROJECTION

We then explore different strategies for the learnable parameter `coef`—which controls the magnitude of the distributional projection added at each layer—in DIPNet: fixing `coef` to a constant (0.5, 1.0 or 1.2) or the optimal value obtained from a learnable run ("Fixed-Learned"), or treating `coef` as a trainable parameter initialized randomly ("Learnable"). In the "Fixed-Learned" setting, we initialize `coef` to the value obtained from "Learnable" setting (1.4121 for ViT-Tiny, -1.4141 for ViT-Small, and 1.4141 for ViT-Base). As Table 13b shows, fixing `coef` to a small constant produces an accuracy comparable to that of the "Learnable" setting, even though the "Learnable" setting itself starts from a random small value and grows to a larger optimum. However, choosing a larger fixed `coef` leads to a clear drop in performance, even when that constant remains below the final magnitude learned by the "Learnable" setting. This contrast highlights the advantage of a learnable `coef`, which can flexibly adjust perturbation strength to the ideal level for all input layers.

### D.7.3 ABLATION STUDY ON LEARNABLE DISTRIBUTIONAL INPUT PROJECTION ON NOISE-LEVEL: GAUSSIAN CORRUPTION EFFECTS ACROSS MODEL SIZES

To quantify the impact of our DIPNet module under varying gaussian noise levels, we evaluate both ViT-Base and ViT-Tiny corrupted by additive Gaussian noise with standard deviation $\sigma \in \{0.2, 0.5, 1.0\}$. Table 14 reports test accuracy for the standard (baseline) training and for our method.

---

[6] https://github.com/jeonsworld/ViT-pytorch

Table 12: Accuracy (%) of ViT backbones trained from scratch on CIFAR-100.

| Model | Standard | $(\alpha, \beta) = (0.1, 0.1)$ | | | $(\alpha, \beta) = (0.1, 0.2)$ | | | $(\alpha, \beta) = (0.1, 0.5)$ | | |
|---|---|---|---|---|---|---|---|---|---|---|
| | | $\lambda = 0$ | $\lambda = 0.05$ | $\lambda = 0.1$ | $\lambda = 0$ | $\lambda = 0.05$ | $\lambda = 0.1$ | $\lambda = 0$ | $\lambda = 0.05$ | $\lambda = 0.1$ |
| Small 16 | 38.28 | 37.91 | **39.42** | 38.53 | 34.87 | 38.77 | 38.79 | 38.69 | 37.75 | 38.08 |
| Small 32 | 31.57 | 31.16 | 31.85 | 31.74 | 31.10 | 30.54 | 31.79 | 31.06 | **32.97** | 30.65 |
| Base 16 | 38.71 | 38.94 | 39.51 | 33.79 | 36.42 | 36.72 | 37.16 | 36.74 | 36.28 | **40.58** |
| Base 32 | 31.69 | 31.10 | 30.75 | 25.42 | 30.43 | **32.80** | 25.47 | 30.42 | 29.73 | 30.05 |

*Notes. Model* refers to backbone size + patch size (e.g., "Base 32" denotes ViT-Base with patch size 32). **Bold** indicates the best performance, and underlined indicates the second best.

Table 13: **Ablation studies on ViT under Gaussian attack.** This table reports test accuracy (%) on the CIFAR-100 dataset. "Standard" is the baseline ViT without any distributional input projection; "Full-Layer" (in a) and "Learnable" (in b) are the same learnable full distributional projection-at-every-layer setting.

(a) Depth Ablation (Layerwise DIPNet).

| Method | ViT-Tiny | ViT-Small | ViT-Base |
|---|---|---|---|
| Standard | 46.31 | 75.65 | 69.13 |
| 1-Layer | 50.39 | 75.55 | 68.81 |
| 2-Layer | **51.90** | 75.99 | **71.78** |
| Full-Layer | 51.62 | **78.21** | 69.31 |

(b) Perturbation Coefficient Ablation (Learnable vs. Fixed DIPNet).

| Method | ViT-Tiny | ViT-Small | ViT-Base |
|---|---|---|---|
| Standard | 46.31 | 75.65 | 69.13 |
| Fixed-0.5 | **51.91** | 77.02 | **69.95** |
| Fixed-1.0 | 48.94 | 75.97 | 67.23 |
| Fixed-1.2 | 47.44 | 75.24 | 65.93 |
| Fixed-Learned | 45.24 | 72.29 | 64.82 |
| Learnable | 51.62 | **78.21** | 69.31 |

For larger model, DIPNet yields consistent gains over the baseline at all noise levels. The gap widens as noise increases, demonstrating that DIPNet effectively enhances robustness to heavy corruption. For smaller model, while DIPNet brings large relative gains at low to moderate noise, the absolute accuracy remains low. This indicates that ViT-Tiny's limited capacity leads to underfitting when faced with strong noise, and suggests that further architectural capacity or noise-specific regularization may be required. These ablation results confirm that DIPNet consistently improves noise robustness, but also highlight the interaction between module efficacy and backbone capacity under severe corruption.

Table 14: Evaluation under Gaussian Noise Levels ($\sigma \in \{0.2, 0.5, 1.0\}$).

| **Model** | **Method** | **0.2** | **0.5** | **1.0** |
|---|---|---|---|---|
| ViT-Base | Standard | 69.13 | 54.81 | 35.02 |
| | DIPNet | **69.23** | **54.95** | **36.08** |
| ViT-Tiny | Standard | 46.31 | 26.99 | 15.93 |
| | DIPNet | **52.22** | **32.75** | **17.63** |

# E LLM USAGE STATEMENT

We used LLMs to aid in polishing the writing of this paper. Specifically, LLMs were employed as a general-purpose assistant to improve clarity, grammar, and style, and to suggest alternative phrasings for technical explanations. They were not used to generate novel research ideas, design experiments,

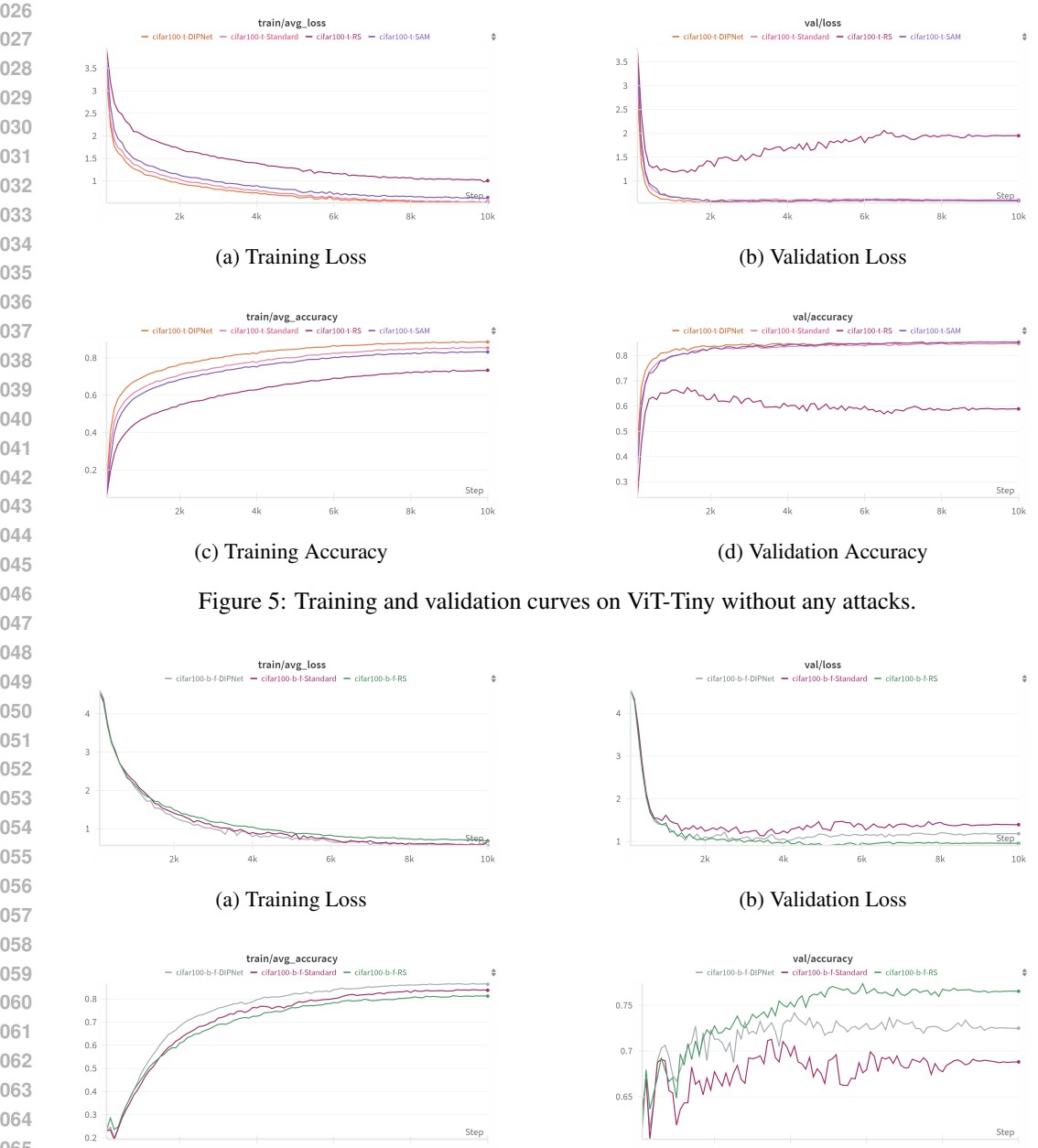

(a) Training Loss

(b) Validation Loss

(c) Training Accuracy

(d) Validation Accuracy

Figure 5: Training and validation curves on ViT-Tiny without any attacks.

(a) Training Loss

(b) Validation Loss

(c) Training Accuracy

(d) Validation Accuracy

Figure 6: Training and validation curves on ViT-Base under FGSM adversarial attacks.

or produce results. The authors take full responsibility for all content, including text refined with the assistance of LLMs.

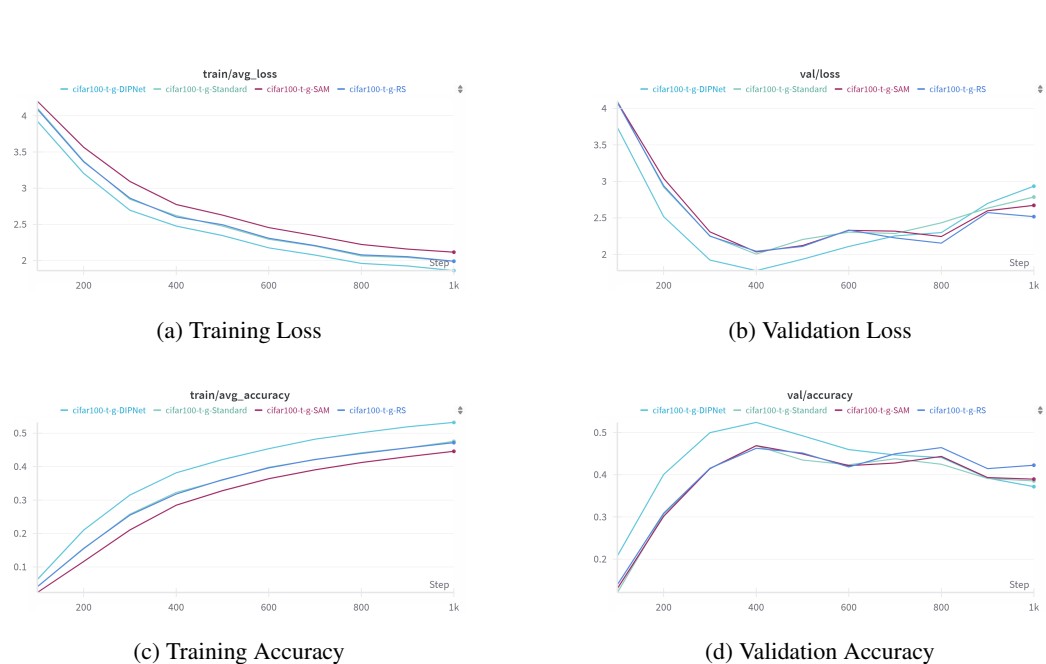

(a) Training Loss

(b) Validation Loss

(c) Training Accuracy

(d) Validation Accuracy

Figure 7: Training and validation curves on ViT-Tiny under Gaussian noise.

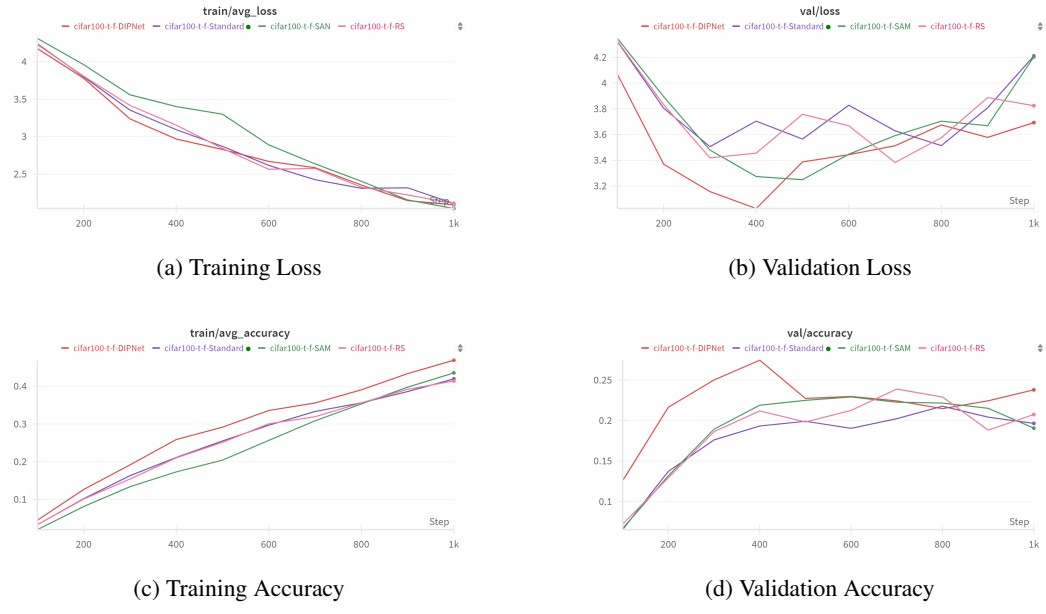

(a) Training Loss

(b) Validation Loss

(c) Training Accuracy

(d) Validation Accuracy

Figure 8: Training and validation curves on ViT-Tiny under FGSM adversarial attacks.

