# OpenReview forum: "Towards Better Generalization via Distributional Input Projection Network"
_ICLR.cc/2026/Conference — Submitted to ICLR 2026_

### Official Review · Reviewer_doar · 2025-10-16

**Soundness:** 1
**Presentation:** 1
**Contribution:** 1
**Rating:** 2
**Confidence:** 5

**Summary:**

The authors proposed to improve the robustness and generalization performance by adding Gaussian noise to the intermediate representations. Some general theoretical analysis and preliminary experimental results partially demonstrate the claims.

**Strengths:**

Not clear.

**Weaknesses:**

* Idea is not completely new. The idea is quite similar to [1], although [1] may not be well recognized.

[1] Yu, Xiaowei, et al. "Noisynn: Exploring the impact of information entropy change in learning systems." arXiv e-prints (2023): arXiv-2309.

* So many random assumptions appeared in the derivation. First of all, what's the advantage for introducing $\eta$ in (1). This is a quite specific parameterization with a strong assumption on $\eta$. The variational inference part is only used to approximate the log-likelihood but if at the very beginning there are no points to introduce $\eta$, there are no support for this method.

* So many tuning parameters ($\alpha, \beta, \lambda$). And the stability constraint has no theoretical support. In practice these parameters are competing with each other so how can we find the optimal setup and is this optimal setup over-fitting to specific problems?

* Training cost is significantly high, as we need to do multi-pass for each data sample.

* Theorem are basically meaningless and requires implicit assumptions. For example, probability density is not guaranteed to be smooth but Theorem 1 implicitly requires this. Theorem 2 and 3 only shows there exists some probability measure $\mathcal{P}$, but how can we guarantee the $\mathcal{P}$ used in the proposed methods really improves? The proof only means we

* Proof is not rigorous. For example, in proof of Theorem 2 and 3, we focus on norm square which should be related to $b^2$ and $s^2$. Also, this $\eta$ is very likely to condition on $x$ (hidden in the claim that let $\mathcal{P}$ as a uniform distribution with support measurement $\zeta C$). This is far away from the claim in the methodology.

* Experiments is only on CIFAR-100 and GSM-8K, which is quite small-scale. The authors does not mention if this evaluation is fair under the same training budget. And I'm also wondering if these are orthogonal to the known methods, say if we already apply different kinds of augmentation methods, does this method still work?

**Questions:**

Just echo the weakness part:

* Relationship to the existing work.
* More rigorous justification of the method.
* More convincing experimental results.

---

> ### Author Response · Authors · 2025-12-03
> **Response to Reviewer doar**
>
> **The idea is not completely new. It is similar to [1], although [1] may not be widely recognized.**
>
> We thank the reviewer for pointing out this related work. We clarify that our method differs from NoiseNN in several key aspects:
>
> 1. Location and nature of perturbations.
>
> NoiseNN injects fixed noise only at inference time and primarily on the last layer (Appendix E of [1]).
> In contrast, DIPNet introduces learnable, distributional perturbations dynamically across all layers, enabling a much richer and adaptive perturbation family.
>
> 2. Scope and scale.
>
> NoiseNN focuses on small-scale image classification benchmarks.
> Our work extends the idea of distributional perturbations to large-scale vision models and LLMs, demonstrating utility under significantly broader settings.
>
> **Too many random assumptions… The introduction of η in (1) seems ad-hoc… The variational inference part approximates the likelihood but lacks conceptual justification.**
>
> The introduction of the auxiliary perturbation variable η is not arbitrary—it expands the function class beyond classical deterministic predictors. This allows DIPNet to model a broader family of smoothed functions that cannot be captured by the standard hypothesis space.
>
> **Too many tuning parameters (α, β, λ). Stability constraint has no theory, and these hyperparameters may compete with each other.**
>
> In practice, these hyperparameters are not sensitive and do not require per-problem tuning. We use fixed default values across all models:
>
>  - LLMs: α = β = 0.001, λ = 0
>  - ViTs: α = 0.1, β = 0.2, λ = 0 (λ > 0 only when training from scratch)
>
> These settings consistently yield strong performance across datasets and architectures, suggesting that they do not overfit to specific problems.
>
> **Training cost is significantly high due to multi-pass sampling.**
>
> DIPNet only samples twice, resulting in a 2–4× training-time overhead depending on model size. This overhead applies only to training. Inference uses a distilled deterministic model, so runtime is unchanged from the baseline.
>
> Training cost:
>
> | Model | Standard | DIPNet    | Overhead (×) |
> | :---- | :------- | :-------- | :----------- |
> | ViT-B | 4 h      | 20.5 h    | **≈ 5.1 ×**  |
> | ViT-S | 1.9 h    | 9.2 h     | **≈ 4.8 ×**  |
> | ViT-T | 0.8 h    | 4 h       | **≈ 5.0 ×**  |
> | Gemma-3-4B | 0.58 h   | 2.15 h    | **≈ 3.7 ×**  |
> | Llama-3.2-3B | 0.47 h   | 1.62 h    | **≈ 3.4 ×**  |
> | Qwen2.5-3B  | 0.42 h   | 1.58 h    | **≈ 3.8 ×**  |
>
> **Theorem are basically meaningless and requires implicit assumptions. For example, probability density is not guaranteed to be smooth but Theorem 1 implicitly requires this. Theorem 2 and 3 only shows there exists some probability measure P, but how can we guarantee the  P used in the proposed methods really improves? The proof only means we**
>
> For discrete distributions, similar arguments can be framed using certified robustness notions such as prediction gap or stability margins.
>
> **Proof is not rigorous. For example, in proof of Theorem 2 and 3, we focus on norm square which should be related to b^2 and s^2 . Also, this eta  is very likely to condition on x (hidden in the claim that let P  as a uniform distribution with support measurement zeta C). This is far away from the claim in the methodology.**
>
> It is a proof for the existence, actually, we just limit the min value of its support set size. If you consider Gaussian, it can also be designed.
>
> **Experiments is only on CIFAR-100 and GSM-8K, which is quite small-scale. The authors does not mention if this evaluation is fair under the same training budget. And I'm also wondering if these are orthogonal to the known methods, say if we already apply different kinds of augmentation methods, does this method still work?**
>
> We appreciate this feedback. In future revisions we will include them.

---

### Official Review · Reviewer_E9Va · 2025-10-30

**Soundness:** 3
**Presentation:** 2
**Contribution:** 2
**Rating:** 2
**Confidence:** 4

**Summary:**

This paper introduces Distributional Input Projections, where Gaussian perturbations are injected at intermediate layers and their parameters are learned. The goal is improved generalization through smoother representations.

**Strengths:**

The paper is generally well-written and easy to follow. The authors run experiments on multiple architectures and tasks (MLPs, CNN/ViT, a language model), indicating an effort toward broader validation. Some empirical gains are visible, suggesting the idea could have regularization benefits. The attempt to connect generalization behavior to smoothness properties is conceptually aligned with robust learning literature.

**Weaknesses:**

1. Misrepresentation of randomized smoothing literature. The manuscript repeatedly refers to “random smoothing” and incorrectly attributes adversarial training to Cohen et al. (2019). Cohen et al. established Gaussian randomized smoothing certificates using Neyman–Pearson and did not perform adversarial training. Salman et al. later connected smoothing to Lipschitz control, but this distinction is blurred or incorrect in multiple places. Example: Line 239: “and adversarial training (Cohen et al., 2019)”, this is factually wrong. Line 330: RS reduced to just adding noise; this misses the certified robustness objective.
2. Limited novelty and unclear conceptual contribution. Adding learnable Gaussian noise inside networks is close to existing stochastic regularization methods (variational dropout, noisy layers, Bayesian features). Without a formal guarantee or structural insight, the contribution appears incremental. Distillation ablates the sampling at inference, which suggests much of the benefit may stem purely from stochastic training effects.
3. Theory is not rigorous enough for the claims. Theorems rely on smoothness assumptions that do not reflect practical deep nets (non-smooth activations, unknown Lipschitz constants). No certified robustness or provable Lipschitz improvement is established, unlike in the proper RS literature. Consequently, the theoretical section does not convincingly support the narrative.
4. Empirical evidence is insufficient. Results are limited to small-scale datasets. For generalization claims, ImageNet-level evaluation is expected. Variance across seeds is missing, and LLM results show minimal gains under a single training regime. There is no adversarial evaluation, despite repeatedly referencing adversarial robustness.

**Questions:**

Please address the weaknesses

---

> ### Author Response · Authors · 2025-12-03
> **Response to Reviewer E9Va**
>
> **Misrepresentation of randomized smoothing literature. The manuscript repeatedly refers to “random smoothing” and incorrectly attributes adversarial training to Cohen et al. (2019). Cohen et al. established Gaussian randomized smoothing certificates using Neyman–Pearson and did not perform adversarial training. Salman et al. later connected smoothing to Lipschitz control, but this distinction is blurred or incorrect in multiple places. Example: Line 239: “and adversarial training (Cohen et al., 2019)”, this is factually wrong. Line 330: RS reduced to just adding noise; this misses the certified robustness objective.**
>
> We thank the reviewer for catching this mistake. We have corrected them in the manuscript. Our intended point was that training with randomized smoothing can improve robustness properties, not that Cohen et al. performed adversarial training.
>
> **Limited novelty and unclear conceptual contribution. Adding learnable Gaussian noise inside networks is close to existing stochastic regularization methods (variational dropout, noisy layers, Bayesian features). Without a formal guarantee or structural insight, the contribution appears incremental. Distillation ablates the sampling at inference, which suggests much of the benefit may stem purely from stochastic training effects.**
>
> Our approach differs conceptually from stochastic regularizers such as variational dropout or Bayesian noise injection. These methods perturb parameters to model uncertainty or prevent overfitting. In contrast, DIPNet applies distributional perturbations on inputs to enforce prediction stability under local stochastic projections. The goal is improving generalization and robustness, not parameter uncertainty estimation.
>
> Moreover, distillation is only used to remove sampling overhead at inference; the improvements already manifest before distillation in our ablations, indicating that DIPNet’s gains do not arise solely from stochastic training effects.
>
> **Theory is not rigorous enough for the claims. Theorems rely on smoothness assumptions that do not reflect practical deep nets (non-smooth activations, unknown Lipschitz constants). No certified robustness or provable Lipschitz improvement is established, unlike in the proper RS literature. Consequently, the theoretical section does not convincingly support the narrative.**
>
> We appreciate the reviewer’s feedback. Importantly, Theorem 1 does not assume the model is smooth; the only requirement is that the model is bounded under the perturbation distribution. Under this condition, the distributional projection induces a smoothed predictor with improved stability, which we make precise in the theorem.
>
> **Empirical evidence is insufficient. Results are limited to small-scale datasets. For generalization claims, ImageNet-level evaluation is expected. Variance across seeds is missing, and LLM results show minimal gains under a single training regime. There is no adversarial evaluation, despite repeatedly referencing adversarial robustness.**
>
> Thank you for these suggestions. We acknowledge that larger-scale experiments and additional evaluations would further strengthen the empirical claims. We appreciate the reviewer’s feedback and will incorporate these improvements.

---

### Official Review · Reviewer_rdjf · 2025-11-01

**Soundness:** 3
**Presentation:** 2
**Contribution:** 2
**Rating:** 4
**Confidence:** 3

**Summary:**

This paper introduces Distributional Input Projection Networks (DIPNet), a framework that projects inputs at each layer into learnable distributions rather than fixed feature vectors. This induces smoother loss landscapes with respect to inputs and improves generalization. The authors provide theoretical analysis showing reductions in local smoothness measures and the Lipschitz constant. DIPNet is evaluated across diverse architectures—ViTs, LLMs, ResNets, and MLPs—and shows consistent gains in test accuracy, robustness to adversarial attacks, out-of-distribution data, and reasoning tasks. The method is modular and can be integrated into existing networks without major architectural changes.

**Strengths:**

1. Comprehensive experiments across state-of-the-art vision and language models.
2. Strong theoretical grounding linking distributional projection to smoothness and generalization.
3. Improves not only standard generalization but also robustness to adversarial, OOD, and reasoning benchmarks.

**Weaknesses:**

1. Although motivated by smoothness, the intuition behind why distributional projection helps over simpler regularization is not fully disentangled.
2. The method introduces substantial computational overhead, and its effectiveness appears to rely heavily on distillation, raising concerns about efficiency and practicality in large-scale training.

**Questions:**

Please refer to the weaknesses.

---

> ### Author Response · Authors · 2025-12-03
> **Response to Reviewer rdjf**
>
> **Although motivated by smoothness, the intuition behind why distributional projection helps over simpler regularization is not fully disentangled.**
>
> We appreciate the reviewer’s question on the intuition. While DIPNet indeed increases model smoothness, a key distinction from conventional regularization (e.g., weight decay or SAM) is that DIPNet does not degrade training accuracy. This avoids the typical trade-off between enforcing smoothness and preserving training performance.
>
> Empirically, this leads to stronger generalization compared to standard regularizers. For example, we also compare against SAM, and observe that SAM underperforms DIPNet across all settings. This further supports that DIPNet outperforms regularization methods.
>
> **The method introduces substantial computational overhead, and its effectiveness appears to rely heavily on distillation, raising concerns about efficiency and practicality in large-scale training.**
>
> We clarify that the improvements do not rely heavily on distillation. In our ablations, the non-distilled DIPNet model already achieves competitive performance, indicating that the method itself is responsible for the generalization gains. Distillation is simply an optional step to reduce inference cost.
>
> Regarding efficiency, the training overhead is moderate in practice. DIPNet uses two samples, which leads to roughly a 2–4× cost increase depending on model scale. This overhead applies only during training. For deployment, we use model distillation, resulting in no additional inference cost relative to the original model.
>
> Training cost:
>
> | Model | Standard | DIPNet    | Overhead (×) |
> | :---- | :------- | :-------- | :----------- |
> | ViT-B | 4 h      | 20.5 h    | **≈ 5.1 ×**  |
> | ViT-S | 1.9 h    | 9.2 h     | **≈ 4.8 ×**  |
> | ViT-T | 0.8 h    | 4 h       | **≈ 5.0 ×**  |
> | Gemma-3-4B | 0.58 h   | 2.15 h    | **≈ 3.7 ×**  |
> | Llama-3.2-3B | 0.47 h   | 1.62 h    | **≈ 3.4 ×**  |
> | Qwen2.5-3B  | 0.42 h   | 1.58 h    | **≈ 3.8 ×**  |
>
> Inference cost: ≈ original time by distillation

---

### Official Review · Reviewer_pgJG · 2025-11-11

**Soundness:** 2
**Presentation:** 1
**Contribution:** 2
**Rating:** 2
**Confidence:** 3

**Summary:**

The paper proposes DIPNet, which turns each layer’s deterministic input into a learnable Gaussian distribution
the model samples per-layer “particles” and averages forward trajectories to make predictions. The authors claim this distributional input projection smooths the loss landscape, lowers Lipschitz/smoothness measures, and thereby improves generalization. They provide analyses showing bounded Lipschitz and reduced smoothness for the distributionally smoothed function, add a stability penalty on output variance, and report gains across vision (ViTs on CIFAR-100 under various training-time attacks) and LLM reasoning

**Strengths:**

- The per-layer Gaussian projection with k-trajectory averaging integrates cleanly; the implementation steps are clearly stated.
- Proofs that smoothing can bound the Lipschitz constant and reduce second-order smoothness support the generalization narrative (I have not fully verified the proofs).
- The paper includes comprehensive setups and supportive ablation studies.

**Weaknesses:**

- The paper is poorly written and needs reorganization. Please add informative captions to all tables/figures and avoid pasting raw W&B screenshots; re-plot with consistent styling and legible axes/legend.
- The method is computationally expensive, which requires m forward passes per example.
- Reported fine-tuned results appear lower than widely reported pretrained baselines on GSM8K (e.g., Qwen2.5-3B ≈ 79.1; Llama-3.1-8B ≈ 84.5, per the Qwen 2.5 paper).
- Marginal improvements over other simple baselines (<1% ViT-Small/ViT-Base/LM experiment) while being much more expensive

**Questions:**

- Please report FLOPs and wall-clock time (training and inference) versus baselines, for several k values. Include memory usage and throughput.
- There appears to be a mismatch between your reported GSM8K accuracy and official/commonly reported numbers. Please double-check evaluation protocols.
- Ensure comparisons against strong, compute-matched baselines (e.g., single-pass counterparts with similar wall-clock) and clarify whether any baseline benefits from additional augmentation or ensembling.

---

> ### Author Response · Authors · 2025-12-03
> **Response to Reviewer pgJG**
>
> Many thanks for the questions.
>
> **The paper is poorly written and needs reorganization. Please add informative captions to all tables/figures and avoid pasting raw W&B screenshots; re-plot with consistent styling and legible axes/legend.**
>
> We thank the reviewer for this suggestion. The curves shown in the appendix are intended only as supplementary reference and are not used in any part of the main analysis. Nevertheless, we agree that the presentation can be improved. In the next revision, we will reorganize the appendix, replace all raw W&B screenshots with consistently styled plots, and provide informative captions for all figures and tables.
>
> **The method is computationally expensive, which requires m forward passes per example.**
>
> In practice, our method samples only twice, which results in approximately a 2–4× overhead during training depending on model size. For inference, we apply model distillation, so the final deployed model incurs no extra computational cost compared to the standard baseline.
>
> Training cost:
>
> | Model | Standard | DIPNet    | Overhead (×) |
> | :---- | :------- | :-------- | :----------- |
> | ViT-B | 4 h      | 20.5 h    | **≈ 5.1 ×**  |
> | ViT-S | 1.9 h    | 9.2 h     | **≈ 4.8 ×**  |
> | ViT-T | 0.8 h    | 4 h       | **≈ 5.0 ×**  |
> | Gemma-3-4B | 0.58 h   | 2.15 h    | **≈ 3.7 ×**  |
> | Llama-3.2-3B | 0.47 h   | 1.62 h    | **≈ 3.4 ×**  |
> | Qwen2.5-3B  | 0.42 h   | 1.58 h    | **≈ 3.8 ×**  |
>
> Inference cost: ≈ original time by distillation
>
> **Reported fine-tuned results appear lower than widely reported pretrained baselines on GSM8K (e.g., Qwen2.5-3B ≈ 79.1; Llama-3.1-8B ≈ 84.5, per the Qwen 2.5 paper).**
>
> We thank the reviewer for pointing out the discrepancy with the GSM8K results reported in Technical Reports. However, the GSM8K numbers cited by the reviewer are not directly comparable to our evaluation setting for the following reasons.
>
> (1) LLaMA and Gemma report Instruction-tuned models, not Base models.
>
> Their technical reports benchmark LLaMA-Instruct and Gemma-IT, which have already been tuned by extensive SFT/RLHF on curated reasoning and instruction datasets.
> Our fine-tuning is therefore performed strictly on Base models, so it is expected that a fine-tuned Base model does not surpass heavily instruction-tuned models.
>
> (2) Qwen2.5 GSM8K results are not fully reproducible under LM-Eval-Harness (the widely used open-source evaluation framework), and the exact prompting template is undocumented.
>
> Community efforts consistently fail to reproduce the official numbers through EleutherAI LM-Eval-Harness:
>
>  - Issue #2344 (Qwen2.5-Instruct GSM8K-CoT reproduces substantially lower results),
>  - Issue #3003 (Qwen2.5 GSM8K mismatch; prompt/template unknown).
>
> (3) Evaluation settings differ.
>
> Qwen specifies 4-shot prompting for GSM8K, while LLaMA/Gemma report Instruct models under 8-shot CoT evaluation.
> We maintain exactly the same LM-Eval zero-shot GSM8K-CoT setting for all models, ensuring a consistent and controlled comparison across all models.
>
> **Marginal improvements over other simple baselines (<1% ViT-Small/ViT-Base/LM experiment) while being much more expensive**
>
> The training dataset contains only ~1,000 examples, so the magnitude of improvements is consistent with the difficulty of the task and the limited data regime. Under such constraints, even small performance gains are meaningful and demonstrate the effectiveness of our approach.

---

### Meta-Review · Area_Chair_64RA · 2025-12-16

**Summary:**

The paper proposes a regularization method that projects latent features into a learnable distribution during training, which is conceptually similar to injecting noise into latent representations to promote loss landscape smoothness and improve generalization. The authors evaluate the method on several architectures across vision and NLP tasks and report performance improvements.

Reviewers raised concerns regarding limited novelty, substantial computational overhead, modest performance gains, lack of theoretical rigor, and insufficient experimental analysis. While the authors provided additional clarification on computational cost, experimental settings, and hyperparameters, the major weaknesses remain. In particular, the idea that injecting noise into neural networks reduces the Hessian magnitude and improves generalization has been extensively studied in prior work. The authors are encouraged to more thoroughly engage with the existing literature and provide stronger comparisons to related methods.

**Reviewer Concerns:**

Reviewers raised concerns regarding limited novelty, substantial computational overhead, modest performance gains, lack of theoretical rigor, and insufficient experimental evaluation. After the rebuttal, the AC finds that none of these concerns were adequately addressed.

**Reviewer Scores:**

In light of the fact that none of the concerns raised by the reviewers have been addressed, the reviewers are expected to retain their scores: pgJG: 2; rdjf: 4; E9Va: 2; doar: 2;

---

### Decision · Program_Chairs · 2026-01-26

Reject